# Bioactive Clerodane Diterpenoids from the Leaves of *Casearia coriacea* Vent

**DOI:** 10.3390/molecules28031197

**Published:** 2023-01-25

**Authors:** Allison Ledoux, Carla Hamann, Olivier Bonnet, Kateline Jullien, Joëlle Quetin-Leclercq, Alembert Tchinda, Jacqueline Smadja, Anne Gauvin-Bialecki, Erik Maquoi, Michel Frédérich

**Affiliations:** 1Laboratory of Pharmacognosy, Center of Interdisciplinary Research on Medicines, CIRM, University of Liège, Avenue Hippocrate 15, 4000 Liège, Belgium; 2Laboratory of Biology of Tumor and Development, GIGA/CIRM, University of Liège, Avenue Hippocrate 15, 4000 Liège, Belgium; 3Pharmacognosy Research Group, Louvain Drug Research Institute, LDRI, Université Catholique de Louvain, UCLouvain, Avenue E. Mounier, B1 72.03, B-1200 Brussels, Belgium; 4Laboratoire de Chimie des Substances Naturelles et des Sciences des Aliments, Université de Réunion, Avenue René Cassin 15, BP 7151, 97715 Saint-Denis, La Réunion, France

**Keywords:** *Casearia coriacea*, clerodane diterpenoids, *Plasmodium*, caseamembrin T, corymbulosin I, isocaseamembrin E

## Abstract

*Casearia coriacea* Vent., an endemic plant from the Mascarene Islands, was investigated following its antiplasmodial potentialities highlighted during a previous screening. Three clerodane diterpene compounds were isolated and identified as being responsible for the antiplasmodial activity of the leaves of the plant: caseamembrin T (**1**), corybulosin I (**2**), and isocaseamembrin E (**3**), which exhibited half maximal inhibitory concentrations (IC_50_) of 0.25 to 0.51 µg/mL. These compounds were tested on two other parasites, *Leishmania mexicana mexicana* and *Trypanosoma brucei brucei*, to identify possible selectivity in one of them. Although these products possess both antileishmanial and antitrypanosomal properties, they displayed selectivity for the malaria parasite, with a selectivity index between 6 and 12 regarding antitrypanosomal activity and between 25 and 100 regarding antileishmanial activity. These compounds were tested on three cell lines, breast cancer cells MDA-MB-231, pulmonary adenocarcinoma cells A549, and pancreatic carcinoma cells PANC-1, to evaluate their selectivity towards *Plasmodium*. This has not enabled us to establish selectivity for *Plasmodium*, but has revealed the promising activity of compounds **1**–**3** (IC_50_ < 2 µg/mL), particularly against pancreatic carcinoma cells (IC_50_ < 1 µg/mL). The toxicity of the main compound, caseamembrin T (**1**), was then evaluated on zebrafish embryos to extend our cytotoxicity study to normal, non-cancerous cells. This highlighted the non-negligible toxicity of caseamembrin T (**1**).

## 1. Introduction

Malaria is a vector-borne disease endemic to tropical and subtropical regions, claiming hundreds of millions of clinical cases and hundreds of thousands of deaths yearly. According to the WHO, there were an estimated 247 million cases of malaria worldwide, with 619,000 malaria deaths in 2021 [1]. The parasite responsible for this disease is *Plasmodium* sp., a protozoan eukaryote that has proven to be a fearsome parasite. Their large genome, containing 23 million bases for *Plasmodium falciparum*, their life cycle occurring both in human and mosquitoes, and the strategy it develops to evade host immunity make them particularly difficult to eradicate. All these parameters account for the emergence and spread of resistance to all available treatments. The control and eradication of malaria will therefore require a multitude of approaches, including prevention and treatment. Concerning prevention, the use of mosquito nets remains the most effective way to fight against malaria [1]. Another widely discussed means of prevention today is vaccination with Mosquirix (RTS,S), especially considering that the WHO has recommended its use since October 6th, 2021, for children at risk in Sub-Saharan Africa [2]. Even though it is not the perfect vaccine, an important clinical trial involving nearly 7000 children (5–17 months) in Burkina Faso and Mali demonstrated that the combination of RTS,S with seasonal malaria chemoprevention provides around 90% protective efficacy, contrasting with the administration of the chemoprevention alone that provides around 70% protection (as compared with no intervention) [3,4]. Although vaccination is an additional weapon in the fight against this disease, malaria treatment is an important control measure and still relies heavily on drugs. The treatment of malaria is currently based on eight major classes of compounds: the 4-aminoquinolines in which we find chloroquine, piperaquine and amodiaquine, the 8-aminoquinolines in which we find primaquine and tafenoquine, the antifolates, containing pyrimethamine and proguanil, the sulfonamides, containing dapsone and sulfadoxine, the 4-amino alcohols, containing quinine, mefloquine, halofantrine, lumefantrine, the napthoquinones, containing atovaquone, the antibiotics, containing doxycycline, and the endoperoxides, containing artemisinin, artesunate, artemether, and dihydroartemisinin. However, their efficacy gradually decreases as parasites develop resistance [5]. It is urgently necessary to discover new classes of antimalarial drugs.

As more than 60% of antiparasitic therapeutic agents approved between 1981 and 2019 were unaltered natural products (10%), natural product derivatives (35%), or synthetic drugs with a natural product pharmacophore (15%) [6], and as the potential of plants for the discovery of new antimalarial drugs is widely supported by the literature, research of new compounds in plants remains a rational approach [5,6,7].

Considering the above-mentioned elements, a large screening of 64 endemic plants of the Mascarene Islands was carried out by our team and highlighted the potential of some plants, including *Poupartia borbonica* [8,9,10], *Vernonia fimbrillifera* [11], as well as *Casearia coriacea* [12]. The present study describes the bioassay-guided fractionation of *Casearia coriacea* that was achieved with the *P. falciparum* strain.

The genus *Casearia* (Salicaceae *sensu lato*) is known to contain more than 180 species, which possess a lot of bioactive compounds [13]. These bioactivities include cytotoxic [14,15,16,17], immunomodulatory [18], anti-inflammatory [19] and antiparasitic activities [20]. Previous studies have suggested that clerodane-type diterpenoids are the major active constituents of *Casearia* [21], and therefore, they should be deeply investigated. Concerning their antiprotozoal activities, different *Casearia* species can be highlighted: *C. sylvestris*, *C. elliptica* and *C. grewiifolia. C. sylvestris* contains casearins A, B, J and G, which exhibit antileishmanial and antitrypanosomal activities [20]. The leaf and bark extracts of *C. sylvestris* also showed antiplasmodial activity, with an IC_50_ between 0.9 and 7.7 µg/mL [22]. *C. elliptica* leaves and stem extracts showed antiplasmodial activity with an IC_50_ between 9 and 33 µg/mL [23], and clerodane diterpenes isolated from *C. grewiifolia* bark also exhibit considerable antiplasmodial activity (IC_50_ between 2.4 and 3.3 µg/mL) [24].

*Casearia coriacea* Vent., also called “bois de cabri rouge”, is an endemic shrub from the Mascarene Islands belonging to this genus. As far as it could be established, this plant has never been phytochemically studied. A bioassay-guided fractionation was performed on the dichloromethane leaf extract. Different chromatographic techniques allowed us to obtain three known compounds. These compounds (**1**–**3**) have never been highlighted in *C. coriacea*, nor investigated for antiplasmodial or antitrypanosomal activity.

Biological assays were further performed with the isolated compounds on human breast, pancreas, and lung carcinoma cell lines (MDA-MB-231, PANC-1, and A549 respectively) to assess selectivity between parasites (*Plasmodium*) and human cells. Antiparasitic assays were also conducted on *Leishmania* and *Trypanosoma* to highlight any selectivity between the parasites investigated. To observe the toxicity of these compounds on healthy cells, a zebrafish embryo acute toxicity test was conducted.

Herein, we describe the isolation and purification of the compounds and the evaluation of the selected bioactivities.

## 2. Results and Discussion

Ethyl acetate extracts were prepared from the roots and the leaves of *C. coriacea*. Both demonstrated promising antiplasmodial activities with IC_50_ (half maximal inhibitory concentration) under 5 µg/mL. These results were in agreement with those we obtained during the previous screening, both for the roots and the leaves. The leaves were chosen for the continuation of the investigation, as it is a plant endangered by extinction.

Three different extracting solvents were tested to select the initial point of the bioassay-guided fractionation, and the resulting dry extracts were tested on the *Plasmodium falciparum* 3D7 strain (Table 1).

The CH_2_Cl_2_ crude extract, which exhibits the highest activity, was defatted using a two-solvent-phase system (*n*-hexane/MeOH/CH_3_CN) and both phases were tested on the *P. falciparum* 3D7 strain. The active compounds were more concentrated in the polar phase than in the apolar phase (IC_50_ of 0.33 µg/mL and 9.34 µg/mL, respectively).

The fractionation of the active lower phase performed by Dry Column Vacuum Chromatography yielded 41 fractions grouped according to their TLC profiles into 5 fractions, **A**–**E**.

The fraction B was the most active. The preparative HPLC performed on fraction B resulted in the purification of three compounds, **1**–**3**. These compounds were identified as clerodane diterpenes by NMR and MS techniques (Figure 1). Previous studies reported that clerodane diterpenes from the *Casearia* genus, such as those from *C. grewiifolia* [25], *C. graveolens* [17], *C. rupestris* [26], *C. sylvestris* [27], and many others [13] are the most active constituents of the genus *Casearia*. They display antiplasmodial, antifungal, antimicrobial, and cytotoxic properties.

Compounds **1**–**3** were isolated as colorless compounds. The molecular formula of compound **1** was determined by HR-ESI-MS as C_28_H_40_O_8_ from the molecular ion at *m/z* 527.2629 [M + Na]^+^. The NMR spectra indicated that **1** has three ﻿ester substituents and a clerodane diterpene skeleton, similar to those found in other *Casearia* sp. [17]. Compound **1** was then identified as caseamembrin T through comparison with literature data [16]. Compound **2** possesses the same molecular formula as compound **1** from the ion at *m/z* 527.2612 [M + Na]^+^ and the ^1^H-. The ^13^C-NMR data of compound **2** were similar to those of compound **1**, also indicating a clerodane diterpene, with a different ester group at C-2 consisting of an isobutanoyl ester instead of a butanoyloxy group in compound **1**. Compound **2** was then identified as corymbulosine I through comparison with literature data [28]. The molecular formula of compound **3** was determined as C_29_H_38_O_8,_ from the ion at *m/z* 541.2759 [M + Na]^+^, and identified as rel-(2S,5R,6R,8S,9S,10R,18S,19R)-18,19-epoxy-18,19-diacetoxy-6-hydroxy-2-(2-methylbutanoyloxy)cleroda-3,13(16),14-triene, a diastereomer of caseamembrin E through comparison with the NMR literature data, as caseamembrin E revealed a relative deshielding of C-2 (δ 70.5) compared to C-2 of **3** (δ 66.6). This slightly shielded C-2 signal, compared to caseamembrin E, is typical of a α-orientation of H-2 [14,29,30]. This compound was renamed isocaseamembrin E. The ^13^C-NMR data information of compounds **1**–**3**, their proton spectra, the comparison with literature, and their proton and carbon assignments are available as Appendix A.

Caseamembrin T (**1**) was previously isolated from the roots of *Casearia membranacea* Hance., a tropical tree growing in Taiwan. Ching-Yu Chen et al. demonstrated its cytotoxicity on different cancer cell lines [16]. Corymbulosin I (**2**) was previously extracted from the bark of *Leatia corymbulosa*, collected in Peru and belonging to the Salicaceae family [28]. Corymbulosin I demonstrated cytotoxic activities against cancer cell lines [28]. Isocaseamembrin E (**3**) is a cytotoxic compound first identified in the bark of *Casearia tremula* [29] (collected in Costa Rica), then in the bark of *C. lucida* [30] (collected in Madagascar), and in the leaves and twigs of *C. membranacea* [14] (collected in Iceland), and more recently in its roots [16].

According to the results described in Table 2, compounds **1**–**3** demonstrated very high antiplasmodial potentiality, particularly caseamembrin T with an IC_50_ of 0.25 ± 0.10 µg/mL. This is the first time that these compounds have been shown to possess antiparasitic activities and have been identified in this plant. Results demonstrated that there is more activity against *Plasmodium*, an intracellular parasite, than against *Leishmania* and *Trypanosoma*. The conjugation of carbonyl and olefinic groups in compounds **1**–**3** makes them Michael acceptors, which could partially explain their antimicrobial activities [31]. These results also demonstrated that compounds **1**–**3** were more active against *Trypanosoma* than against *Leishmania*. A study performed by Dinis Bou et al. [20] also established that some clerodane diterpenes (casearins A, B) demonstrated higher efficacy against *Trypanosoma* than against *Leishmania*, and this may be explained by their stronger interaction with the plasma membrane of these parasites, compared to the *Leishmanias*’ plasma membrane [20]. In this context, we can assume that our compounds behave in the same way. In such a case, the affinity for the plasma membrane of *Plasmodium* would be even more important. It should also be mentioned that it is necessary for the compounds to cross upstream the red blood cell membrane. Therefore, cytotoxicity assays were performed to evaluate if these compounds act more specifically on mammalian cells.

Compounds **1**–**3** were as active on mammalian cells (Table 3) as on *P. falciparum* (Table 2), with a better activity for compound **1** towards *Plasmodium*, and a selectivity index against MDA-MB-231, PANC-1 and A549 cell lines of 5.4, 3, and 6.5, respectively. The IC_50_ expressed in µg/mL and in µM are available as Appendix A. The importance of the biological activity of these compounds could be conditioned by their capacity to penetrate the plasma membrane and enter the cells, where they can interact with the DNA, as Michael acceptors are well described as DNA-interacting molecules [32].

Compounds **1**–**3** exhibited slightly selective cytotoxic activity for the pancreatic carcinoma (PANC-1) cell line (Table 3). Corymbulosin I (**2**) and isocaseamembrin E (**3**) were previously tested against MDA-MB-231 and A549 cells by Aimati et al. [28]. The concordance of our results obtained on identical molecules isolated from different plants allows us to validate our respective studies. Corymbulosin I (**2**) is very promising for its cytotoxic properties. It is not the first time that corymbulosins, clerodane diterpenes with an isozuelanin skeleton, exhibit this kind of activity, encouraging a deeper understanding of the mechanisms of action. Corymbylosin A, X, and M were highlighted in a recent review concerning the potential anti-cancer activities of diterpenes with abietane, labdane, and clerodane skeletons [33]. Corymbulosin X, isolated from *Anacolasa clarkii* exhibits high activity against some pediatric cancer cell lines [34]. Corymbulosin A and corymbulosin M, isolated from *Laetia corymbulosa* and from *Casearia kurzii*, respectively, have also displayed some interest against different cancer cell lines. Corymbulosin A was more active than etoposide (positive control) and induced apoptotic death, which seems to be related to the ability of the compound to arrest the cell cycle at the G0/G1 stage [35]. In the same way, some compounds with very similar structures isolated from another *Casearia*, *C. graveolens*, have been shown to induce cell death through apoptosis [17].

To extend our cytotoxicity study to normal, non-cancerous cells, the activity of caseamembrin T (**1**) was also tested in a *zebrafish* embryo acute toxicity assay. Zebrafish represent a useful in vivo model for assessing compound toxicity. These vertebrates share similarities with non-human mammals in terms of physiology, development, and metabolic pathways, and they exhibit ex utero embryonic development. These embryos can be generated rapidly and are transparent, allowing clear visualization of the effects induced by a compound [36]. Saukat et al. [37] compared the toxicity results of sixty drugs on zebrafish and rodents, in order to verify the predictive power of the zebrafish model. The conclusion is that there is a high overall predictive power, but that the influence of the phytochemical class of the tested molecules is important. For example, carboxylic acids, glycosides, and alkaloids are more toxic for zebrafish than rodents in general, while the opposite is observed for alcohols [37].

Considering compound **1**, neither mortality nor morphological change were observed up to a concentration of 1 µg/mL. At a dose of 5 µg/mL, mortality was observed, 20% of the embryos were alive at 48 hpf but were lifeless at 72 hpf. Concentrations higher than 5 µg/mL were lethal for all embryos (Figure 2). For illustration, as reported by Saukat et al. [37], digoxin has a lethal dose 50 (LC_50_) of 0.5 ± 0.006 µg/mL, nicotine of 35.1 ± 0.5 µg/mL and paracetamol of 535.8 ± 17.1 µg/mL. With a LC50 value in the range of 1 and 5 µg/mL, these results confirmed the important toxicity of compound **1**.

This is the first time that caseamembrin T, corymbulosin I, and isocaseamembrin E have been isolated from *C. coriacea* and evaluated for their antiparasitic activities. Their activities against PANC-1 cells (compounds **1**–**3**), A549 and MDA-MB-231 cells (compound **1**) are also demonstrated for the first time. The possibility of using such compounds as anti-malarials without selective targeting appears to be compromised by their broad toxicity against eukaryotic cells. Nevertheless, it could be interesting to deeply investigate the potentialities of clerodane diterpenes, and more specifically corymbulosin I, for the targeting of pancreatic cancer cells. It would be interesting to perform real-time imaging tests on a pool of cells, which would contain both tumor cells (as PANC-1) and healthy cells (as fibroblasts), in order to have a detailed morphological analysis of the cells when they are put in contact with corymbulosin I, and to try to distinguish a difference in terms of mode of action.

## 3. Experimental Section

### 3.1. General Experimental Procedures

The FT-IR spectra were measured on a Frontier PerkinElmer spectrophotometer equipped with an ATR module. NMR spectra were recorded in CDCl_3_ on a Bruker AVANCE NEO 500 MHz spectrometer equipped with a cryoprobe. MS data were obtained on a 9.4 tesla FTICR mass spectrometer (Bruker Daltonic, Billerica, MA, USA) in positive ion mode. Analytical TLC was performed on precoated Si gel 60 F254 (Merck, Hohenbrunn, Germany) plates. After the development (EtOAc, *n*-hexane, isopropanol [10:9:1]), the dried plates were observed under 254 nm, and sprayed with sulfuric vanillin and heated for 15 min at 110 °C to analyze the extract, the fractions, and the isolated compounds. A Reverse FLASH/PREP chromatography equipped with a Büchi fraction collector was used to perform the preparative separation. The columns used were Luna PFP 5 µm column (250 × 4.60 mm, 100 A) phenomenex (HPLC), and Luna PFP 5 µm column (250 × 30 mm, 100 Å) phenomenex (prepHPLC).

### 3.2. Plant Material

The leaves and the bark of *C. coriacea* were collected on Reunion Island and e identified by Jacques Fournel (Université de la Réunion). Voucher specimens (no.TCN-P103L for the leaves and TCN-102B for the barks) were deposited in the Herbarium of the University of Reunion Island. The leaves and the bark were air-dried at room temperature without sunlight, and pulverized using an electric grinder.

### 3.3. Extraction and Isolation

Ethanol, methanol, and dichloromethane extracts were prepared as follows to select the most promising antiplasmodial extract to be investigated: Dried leaves (20 g) were extracted with ethanol, methanol, or dichloromethane solvents (200 mL) by ultrasonication at room temperature for 30 min. After filtration, the solvent was removed under vacuum, resulting in dried extracts.

Antiplasmodial assays were performed on the extract as described in the section “in vitro antiparasitic activities”. As the dichloromethane dried extract was selected for the bioassay-guided fractionation, dried leaves (100 g) were extracted with CH_2_Cl_2_ (1000 mL) by ultrasonication at room temperature for 30 min. After filtration, the solvent was removed under vacuum. The resulting dry extract (11 g) was defatted using a two-solvent-phase system (*n*-hexane/MeOH/CH_3_CN [6:0.5:3.5]). The resulting dry extract of the biologically active lower phase (1.5 g) was dispersed with celite, and subjected to a Dry Column Vacuum Chromatography (DCVC) (LiChroprep Si 60), eluting with a mobile phase containing EtOAc, *n*-hexane, isopropanol [10:9:1]. Twenty-five fractions were collected and grouped into 6 fractions (A to F) on the basis of their TLC profiles. The biologically active fraction B (120 mg) exhibits one single spot on TLC but reveals three different peaks with HPLC. The flow rate was 1 mL/min with a binary solvent system of formic acid 0.1% in water and MeOH (38:62 to 14:86 v/v in 30 min). Compounds **1**–**3** were eluted between 30 and 40 min. The same separation system was applied using the HPLC preparative system and led to the isolation of compounds **1** (11.04 mg), **2** (28.4 mg), and **3** (5.2 mg). Although compound **2** appears to be the dominant compound based on the extraction yield, the HPLC chromatographic profile obtained indicates that compound **1** is the predominant compound, compared to compounds **2** and **3** (Appendix A, Appendix A). The difference results from the loss related to the extraction process.

### 3.4. In Vitro Antiparasitic Activities

A continuous in vitro culture of asexual erythrocyte stages of *P. falciparum*, chloroquine-sensitive strain 3D7 (ATCC), was maintained following the procedure of Trager and Jensen [38]. Antiplasmodial activity tests were conducted as previously described by the determination of lactate deshydrogenase activity [8]. IC_50_ was calculated from graphs (GraphPad Prism). *T. brucei* strain 427 (Molteno Institute in Cambridge, UK) and *L. mexicana mexicana* (MHOM/BZ/84/BEL46 from Pr. P.A.M Michels of De Duve Institue, UCLouvain) were maintained following the procedure previously described by the team of Pr. Quetin-Leclercq [39]. Antileshmanial and antitrypanosomal assays were conducted as previously described, using the AlamarBlue assay [39]. Artemisinin (purity > 98%, Sigma-Aldrich, Machelen, Belgium), isethionate salt (antileishmanial drug, 98% purity, Sigma-Aldrich, Machelen, Belgium), and suramin sodium salt (antitrypanosomal commercial drug, >99% purity, Sigma-Aldrich Machelen, Belgium) were used as positive controls (for antiplasmodial, antileishmanial, and antitrypanosomal activities, respectively).

### 3.5. In Vitro Cytotoxic Activity

Assays were performed on different human cancer cell lines (MDA-MB-231, PANC-1, and A549 cells obtained from ATCC) to evaluate the cytotoxicity potential of the compounds **1**–**3,** and the crude extract. Cells were grown in high-glucose Dulbecco’s modified Eagle Medium (DMEM, Gibco) supplemented with glutamine (2 mM), penicillin (50 IU/mL), streptomycin (0.5 mg/mL), sodium pyruvate (0.5 mM), and 10% FBS (Foetal Bovine Serum) at 37 °C with 5% CO_2_ in the air. Compounds were tested in 96-well microplates using the Prestoblue^®^ colorimetric assay for cell viability based on the reduction of resazurin in resorufin as previously described [21]. Briefly, 2500 cells were seeded per well in 100 µL of medium supplemented with adequate concentrations of the tested drugs. After 72 h of drug exposure, 75 µL of diluted solution of Prestoblue reagent were added to each well, and the plate was incubated in the dark for 2 h at 37 °C. Any changes in cell viability were detected using fluorescence spectroscopy. The fluorescence of resorufin was measured using a SpectraMax i3 plate reader (Molecular Devices™) with excitation and emission wavelengths set at 560 nm and 590 nm, respectively. The cell viability was expressed as a percentage relative to control-vehicle-treated cells (DMSO 0.1%). IC_50_ values were calculated from graphs based on a regression line in which the resorufin fluorescence values were plotted against the logarithm of drug concentration.

### 3.6. Zebrafish Embryos Acute Toxicity Test

Adult zebrafish (*Danio rerio*) were maintained, fulfilling the criteria of the Ethical Committee for the Use of Laboratory Animals at the University of Liège. They were maintained at 28 °C over a period of 14 h day/10 h night, and fertilized eggs were collected, washed with sterile water, and placed in Petri dishes. The embryos were collected, and their chorions were not removed at 24 hpf. Compound **1** was dissolved in DMSO (the maximum final concentration of DMSO was 0.4%). Twenty embryos were used per condition in a 6-well plate. Each well contained 5 mL of the treatment dose (compound **1** or DMSO 0.4%), which was replaced daily for three days. The embryos were observed under a binocular microscope daily until 72 hpf (hours post fertilization).

### 3.7. Statistical Analysis

The results are expressed as mean ± SD. Statistical significance between the compounds was analyzed on GraphPad Prism 6. IC_50_ (half maximal inhibitory concentration) values were obtained from the graphs.

### 3.8. Illustrations

Illustrations used in the graphical abstract were obtained from Servier Medical Art website.

## Figures and Tables

**Figure 1 molecules-28-01197-f001:**
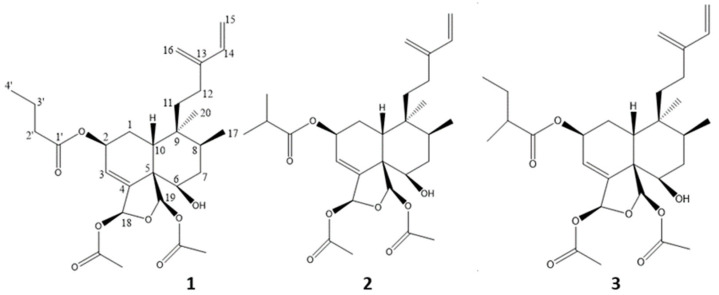
Structures of compounds **1**–**3**.

**Figure 2 molecules-28-01197-f002:**
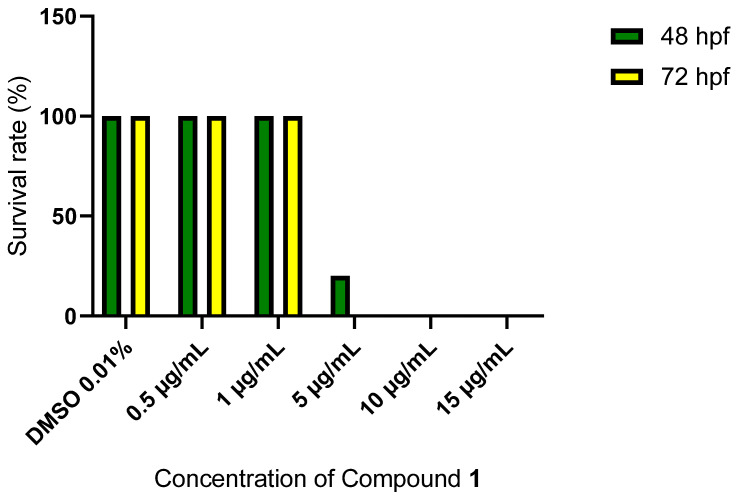
Survival (%) of the zebrafish at 48 (green), and 72 (yellow) hours post fertilization (hpf), exposed to the vehicle (DMSO 0.4%), and Compound **1** (0.5 to 15 µg/mL). *n* = 20 embryo/treatment.

**Table 1 molecules-28-01197-t001:** The table showed the yields obtained for the extractions by the different solvents (mass of extract obtained in relation to the mass of *C. coriacea* leaf powder used to perform the extract, expressed in %), as well as the concentrations that inhibit 50% (IC_50_) of the *P. falciparum* 3D7 parasite growth (performed three times, expressed as mean ± standard deviation, SD). The extract with the best performance and the most interesting activity is the dichloromethane extract.

Extracting Solvent	Yield (%m/m)	IC_50_ (µg/mL) ± SD (*n* = 3)
EtOH	4.19	0.70 ± 0.12
MeOH	4.03	1.89 ± 0.05
CH_2_Cl_2_	4.32	0.62 ± 0.07

**Table 2 molecules-28-01197-t002:** IC_50_ of compounds **1**–**3** and the crude extract on the investigated parasites, expressed in µg/mL ± SD and in µM ± SD). The activities against *Trypanosoma* and *Plasmodium* differ from the anti-leishmanial activity, and the activities of compounds **1** and **3** are closer than those of compound **2**. No product or extract caused hemolytic activity.

Sample	*P. falciparum* 3D7 IC_50_ (*n* = 3) (µg/mL) (µM)	*Leishmania* IC_50_ (*n* = 3) (µg/mL) (µM)	*Trypanosoma* IC_50_ (*n* = 3) (µg/mL) (µM)	Hemolysis (%)
DCM extract	0.62 ± 0.07 -	NT	NT	<1%
Caseamembrin T (1)	0.25 ± 0.10 0.49 ± 0.19	26.05 ± 0.64 51.69 ± 1.27	3.08 ± 0.33 6.11 ± 0.65	<1%
Corymbulosin I (2)	0.40 ± 0.13 0.79 ± 0.26	10.27 ± 0.23 20.38 ± 0.46	3.00 ± 0.60 5.95 ± 1.19	<1%
Isocaseamembrin E (3)	0.51 ± 0.13 0.98 ± 0.25	26.04 ± 0.95 50.27 ± 1.83	3.03 ± 0.49 5.85 ± 0.95	<1%
Artemisinin	0.004 ± 0.001 0.014 ± 0.003	-	-	-
Triton 20%	-	-	-	100%
Pentamidine	-	0.02 ± 0.00 0.06 ± 0.00		-
Suramine		-	0.05 ± 0.008 0.04 ± 0.008	-

**Table 3 molecules-28-01197-t003:** IC_50_ of compounds **1**–**3** and the crude extract on the investigated cell lines, expressed in µg/mL ± SD and µM ± SD. The activity against PANC-1 cells differs from the other two cell lines, and the activity of compounds **1** and **3** are also closer to those of the extract and compound **2**.

Sample	MDA-MB-231 IC_50_ (*n* = 3) (µg/mL) (µM)	PANC-1 IC_50_ (*n* = 3) (µg/mL) (µM)	A549 IC_50_ (*n* = 3) (µg/mL) (µM)
DCM extract	0.89 ± 0.15 -	0.59 ± 0.04 -	0.95 ± 0.12 -
Caseamembrin T (**1**)	1.34 ± 0.01 2.66 ± 0.02	0.74 ± 0.08 1.47 ± 0.16	1.62 ± 0.18 3.21 ± 0.36
Corymbulosin I (**2**)	0.47 ± 0.02 0.93 ± 0.04	0.31 ± 0.10 0.62 ± 0.19	0.75 ± 0.07 1.49 ± 0.14
Isocaseamembrin E (**3**)	1.05 ± 0.25 2.03 ± 0.48	0.61 ± 0.08 1.18 ± 0.15	1.50 ± 0.08 2.89 ± 0.15

## Data Availability

Data is contained within the article or Appendix A.

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
