# Peer review of "Bioactive Clerodane Diterpenoids from the Leaves of Casearia coriacea Vent"

_molecules, 2023, doi:10.3390/molecules28031197_

Round 1

Reviewer 1 Report

The comments:

The abstract should be rephrased and include the purpose of the experiment

Conclusions must be written at the end of the Abstract.

in line 26 you wrote zebrafish larva. But the results are all from the Zebrafish embryo.

The literature review is very poor, many citations are inappropriate and the support of the mechanistic part of the discussion is missing.

In line 61, the scientific names should be: italicizes, C. coriacea and P. falciparum, and the 3D7 strain should be added.

Write an explanation below the table for (%m/m), IC50, and SD

On line 79, put a space at the beginning of the sentence.

In Apragravene from line 130 to line 143 only two references are cited, it is preferable to cite more than that.

Figure 4 The results of this figure are not discussed as well as hpf. What do they mean? Write down.

Figures 2, 3, and 4, Need to discuss and cite recent references.

References are few and most of them are outdated.

Reference 23 is not found in the Manuscript.

Author Response

Dear reviewer,

Thank you for giving me the opportunity to submit a revised version (please see the attachment) of my manuscript entitled «Bioactive clerodane Diterpenoids from the leaves of Casearia coriacea Vent. » submitted to the journal Molecules.

We are grateful to the editor for its patience. Changes are highlighted in yellow in the manuscript. 

Here is a point-by-point response to the reviewers’ comments and concerns.

Reviewer 1

The abstract should be rephrased and include the purpose of the experiment

Conclusions must be written at the end of the Abstract.

--> Abstract was rewritten thanks to the suggestions.

in line 26 you wrote zebrafish larva. But the results are all from the Zebrafish embryo.

--> Modified.

The literature review is very poor, many citations are inappropriate and the support of the mechanistic part of the discussion is missing.

-->Modified in the text.

In line 61, the scientific names should be: italicizes, C. coriacea and P. falciparum, and the 3D7 strain should be added, Write an explanation below the table for (%m/m), IC50, and SD-On line 79, put a space at the beginning of the sentence.

--> Modified.

In Apragravene from line 130 to line 143 only two references are cited, it is preferable to cite more than that.

-->Thank you for the suggestion that allowed us to highlight items that we had not mentioned and that enhance the discussion.

Figure 4 The results of this figure are not discussed as well as hpf. What do they mean? Write down.

Figures 2, 3, and 4, Need to discuss and cite recent references.

--> Modified in the text.

References are few and most of them are outdated.

--> More current references have been added, especially for the part related to cytotoxicity. Some older ones are still kept, because they allowed us to identify with certainty the structure of the compounds.

Reference 23 is not found in the Manuscript.

--> It was a duplicate, thank you for your insight.

Reviewer 2 Report

1. It is very interesting to find clerodane diterpenoids with antiplasmodial effects from an endemic plant Casearia coriacea Vent. However, the logical order of the abstract is confused, and it needs to be rewritten according to the order of experiment advancement. In addition, the end of the abstract needs to be summarized.

2. There are few reports about this plant Casearia coriacea. Please provide more evidence of this plant identification and pictures of plants.

3. Please provide the methods for obtaining methanol and ethanol extracts.

4. The chemical structures of the three compounds in Figure 1 are not clear.

5. The heat maps in Figures 2 and 3 do not reflect the variance significance of the data.

6. Please provide the variance of the data in Figure 4.

Author Response

Dear reviewer,

Thank you for giving me the opportunity to submit a revised version of my manuscript (in attached) entitled «Bioactive clerodane Diterpenoids from the leaves of Casearia coriacea Vent. » submitted to the journal Molecules.

We are grateful to the editor for its patience. Changes are highlighted in yellow in the manuscript. 

Here is a point-by-point response to the reviewers’ comments and concerns.

It is very interesting to find clerodane diterpenoids with antiplasmodial effects from an endemic plant Casearia coriacea Vent. However, the logical order of the abstract is confused, and it needs to be rewritten according to the order of experiment advancement. In addition, the end of the abstract needs to be summarized.

--> Abstract was rewritten thanks to your suggestions.

There are few reports about this plant Casearia coriacea. Please provide more evidence of this plant identification and pictures of plants.

-->This plant is listed in the "world flora online" website and also in the "plant name index". A voucher specimen has been deposited at the herbarium of the University of Reunion n°TCN-P103 (https://collections-umr-pvbmt.cirad.fr/page/Herbarium%20specimens)  and compared with the reference: https://powo.science.kew.org/taxon/urn:lsid:ipni.org:names:779534-1/images

Please provide the methods for obtaining methanol and ethanol extracts.

--> Added in the manuscript

The chemical structures of the three compounds in Figure 1 are not clear.

--> Modified

The heat maps in Figures 2 and 3 do not reflect the variance significance of the data.

-->Standard deviations are indicated on the figure for more transparency, and the table ± SD is available as SI. However, it is true that it is impossible to show this information with a heat map. However, this figure is kept for its visual contribution for the reader, a table not providing more information.

 Please provide the variance of the data in Figure 4.

-->It is not feasible to add a variance for these data, since the larvae are either dead or alive. Only the number of live larvae (out of the 20 placed per well) constitutes a graphical data. In consideration of the toxicity of the compound and the ethical reasons for doing so, we deliberately chose to do the experiment only once, as the additional information (i.e. the standard deviation), does not justify the consumption of 120 additional larvae (20 larvae times the 6 conditions).

Reviewer 3 Report

In the manuscript entitled “Bioactive clerodane Diterpenoids from the leaves of Casearia coriacea Vent.” The authors extract three diterpenes from Casearia coriacea Vent. and demonstrated their bioactivities. The study highlight novelty, but there are some suggestions to improve the manuscript.

1.     In approximately all manuscript the IC50 was written without determining if it was an IC50 value or an IC50 concentration. It should be determined in the whole manuscript.   

2.      The quality of figure 1 is very poor. Please use Molecules guide for authors to improve it.

3.      The legend of figure 1 indicate 1-3 compounds; however, figure 1 represent a, b, c compounds.

4.    In the legend of figure 4, the abbreviation “hpf” should be defined.

5.    In figure 2, Leishmania, Trypanosoma and P.falciparum 3D7 were represented vertical →   represent them horizontally.

6.       In figure 3, A549, MDA-MB-231 and PANC-1 were represented vertical → represent them horizontally.

7.      p. 1, line 36; studies suggested → studies have suggested

8.      p. 1, line 40; screening reveals → screening revealed

9.      p. 2, lines 83, 85, 86 and p. 3, lines 88, 89, 91; 1 → compound 1

10.    p. 3, line 89; data of 2 → data of compound 2

11.    p. 7, line 256; are isolated → were isolated

Author Response

Dear reviewer,

Thank you for giving me the opportunity to submit a revised version of my manuscript (in attached) entitled «Bioactive clerodane Diterpenoids from the leaves of Casearia coriacea Vent. » submitted to the journal Molecules.

In approximately all manuscript the IC50 was written without determining if it was an IC50 value or an IC50 concentration. It should be determined in the whole manuscript.   

-->Added in the text (results & discussion), and in the experimental section.

The quality of figure 1 is very poor. Please use Molecules guide for authors to improve it.

-->Modified with chemdraw

The legend of figure 1 indicate 1-3 compounds; however, figure 1 represent a, b, c compounds.

--> Modified

In the legend of figure 4, the abbreviation “hpf” should be defined.

-->Modified

In figure 2, Leishmania, Trypanosoma and P.falciparum 3D7 were represented vertical →   represent them horizontally.

In figure 3, A549, MDA-MB-231 and PANC-1 were represented vertical → represent them horizontally.

--> Modified

  1. p. 1, line 36; studies suggested → studies have suggested

--> Modified

  1. p. 1, line 40; screening reveals → screening revealed

--> Modified

  1. p. 2, lines 83, 85, 86 and p. 3, lines 88, 89, 91; 1 → compound 1

-->Modified

  1. p. 3, line 89; data of 2 → data of compound 2

--> Modified

  1. p. 7, line 256; are isolated → were isolated

-->Modified by ‘have been isolated’

Round 2

Reviewer 1 Report

I thank the researcher for his effort in improving the manuscript

I think he implemented all the suggestions

best regards

Author Response

Thank you for your time in improving this manuscript.

Kind regards,

A. LEDOUX

Reviewer 2 Report

Because there are only three known compounds in the manuscript, from the perspective of novelty, it is necessary to add reliable activity data to improve the quality of the manuscript. Heat map is an effective method to reflect big data, but it can not directly display SD. In my opinion, the heat map in Figure 2 and Figure 3 cannot be simply used for the activity display of IC50 value. The concentrations setting in the Figure 4 cannot reflect the real activity. The same SD value is also very important.

Author Response

Thank you for your time in improving this manuscript.

I understand that heat map can not directly display SD, I have removed them in order to insert more conventional tables, which better reflect the variations. 

Kind regards,

A. LEDOUX